# Efficient Route Management Method for Mobile Nodes in 6TiSCH Network

**DOI:** 10.3390/s21093074

**Published:** 2021-04-28

**Authors:** Min-Jae Kim, Sang-Hwa Chung

**Affiliations:** Department of Information Convergence Engineering, Pusan National University, Busan 46241, Korea; minjae@pusan.ac.kr

**Keywords:** time slotted channel hopping, mobility, RPL, scheduling

## Abstract

The combination of time slotted access and channel hopping technology in IEEE 802.15 TSCH networks enables high reliability and low power operation to meet the stability and real-time requirements of industrial applications. Basically, TSCH and RPL, a routing protocol for TSCH, are proposed for static nodes that generate fewer control messages, so they allow collisions in shared cells when they exchange control messages. In a topology containing mobile nodes, the collision of control messages in a shared cell makes the network difficult to recover quickly. The proposed scheme minimizes the collision of control messages by allocating dedicated control cells to form preferred parent nodes quickly for mobile nodes. We also proposed a method for establishing a fixed route from the root node to the mobile node in RPL to minimize the delay time. Through the simulation using the 6TiSCH simulator, it was confirmed that the performance of the proposed method was approximately 2.5 times better in terms of overhead and resource use, and 33% better in terms of network participation time of mobile nodes compared with existing solutions.

## 1. Introduction

The Internet of Things (IoT) enables users to collect and share information generated by each device, because each device is connected to the Internet and uses it as a new insight [1]. In particular, industrial IoT technology provides a technique that enables workers to easily collect sensor data by connecting to a network with equipment and sensors in industrial fields [2,3]. Meanwhile, with the development of wireless technology, wireless sensor networks have emerged, and protocols for low-power devices are being applied in various applications with the development of technology. In general, in an industrial environment where industrial IoT applications exist, there are many shielding structures, such as metal, and many sensors are dense; therefore, technology that minimizes radio interference or collision is essential.

Consider a situation in which a worker moves and requests data from a root node that cannot be obtained through sensor values between existing facilities. Sensor data between existing facility devices are continuously collected and transmitted to the root node, and workers request additional data from the root node for efficient work. At this time, the root node should be able to provide the data requested by the moving worker in a short time. Therefore, it is necessary that the mobile node stays connected without losing network connectivity when the worker is on the move. To this end, technology that can change the preferred parent for the mobile node with the minimum delay time should be developed. It is also necessary to set a high-speed path for downlink traffic in response to the request from the mobile node to the root node; thus, the data requested by the mobile node can be quickly received by minimizing the delay time.

The wireless sensor network environment is specialized for limited computer functions and low-power devices, and a network with high reliability and low latency is required because of the characteristics of the industrial environment. Following this trend, the IEEE proposed the IEEE 802.15.4e time slotted channel hopping (TSCH) [4] standard. The TSCH network resolves collisions or contentions that may occur during wireless communication through medium access control technology that divides time into time slots. In addition, through the channel-hopping technique, a multipath fading problem can be solved by preventing collisions because of an increase in time slots and a change in frequency. There are several scheduling techniques that determine when each node sends and receives packets on a TSCH network (at a certain time slot) and through which channel, but research on a scheduling technique to support mobility has not been actively conducted. The node that wants to participate in the network initiates the DODAG Information Object (DIO) message cycle by transmitting the DODAG Information Solicitation (DIS) message. The node receiving the DIS message transmits the DIO message, and the node receiving the DIO message starts the network formation.

As the mobile node moves, the mobile node disconnects from the existing parent node and transmits a DIS message through the shared cell to participate in the network, and the node receiving the DIS message transmits a DIO message through the shared cell. In the process of forming a routing path in the Routing Protocol for Low-Power and Lossy Networks (RPL) protocol, both DIS and DIO messages are transmitted and received in a shared cell, and the cell operates on a contention basis. The more nodes that have received the DIS message, the more nodes that send DIO messages, causing more collisions and data loss of control messages in the corresponding shared cell. Therefore, there is a need for a technique that supports mobility by reducing the collision of control messages in a shared cell.

In contrast to the existing scheduling scheme, the proposed scheme allocates a dedicated cell for the control packet to prevent collisions that may occur during the exchange of RPL control packets, thus, reducing the handover delay time and control overhead of the mobile node. In addition, it is necessary to minimize the dropout of packets generated by the mobile node when the communication link with the existing parent node is temporarily released owing to the movement of the mobile node. Therefore, we propose a scheduling scheme that immediately responds to a change in the link quality of a mobile node, delivers a control packet to a neighboring node, quickly selects an alternate parent node, supplements a path, and minimizes the omission of transmitted packets.

The contributions of this study are as follows:In a TSCH network, a method for minimizing the delay time by constructing a static path for downlink traffic from the root node to the mobile node is proposed, while the effect on the schedule between the existing stationary nodes is minimized.In the process of supporting node mobility in a TSCH network, we propose a technique to reduce the control overhead incurred in a contention-based existing shared cell by allocating a dedicated cell for an RPL control message.A technique to quickly select an alternate parent node in response to the change in link quality of the mobile node and supplement the path to minimize the omission of request packets, and to transmit packets generated from the root node without delay.When forming the routing path of the mobile node in the RPL protocol through the dedicated control cell, the possibility of collision of control messages is reduced by omitting the DIS packet propagation process.

The remainder of this paper is organized as follows: In Section 2, we examine the 6TiSCH network stack proposed by the IETF, and the background knowledge and operation process of the IEEE 802.15.4 TSCH, IETF 6top [5], and IETF RPL [6]. Section 3 discusses the participation of nodes in the network, the process of synchronization and topology formation, the problem of the network in which the mobile node exists, and the fast cell allocation scheduling method of the mobile node. Section 4 presents the results of the experiments using the 6TiSCH simulator. Section 5 presents the conclusions.

## 2. Background

### 2.1. 6TiSCH Network Stack

6TiSCH (IPv6 over the TSCH mode of IEEE 802.15.4e TSCH) [7] is a standard for connecting IPv6 and the performance of industrial protocols of the low-power wireless personal area network IEEE 802.15.4e TSCH.

The network stack structure of 6TiSCH is shown in Figure 1, which uses IEEE 802.15.4 PHY [8] and IEEE 802.15.4e TSCH MAC protocols, IETF 6top, which plays the role of scheduling and monitoring IEEE 802.15.4e TSCH, 6LoWPAN [9], which performs header compression, link step fragmentation, and reassembly mechanism for transmitting IPv6 packets to IEEE 802.15.4 frames, and a destination-oriented directed acyclic graph (DODAG) in a wireless sensor network environment [10].

The wireless sensor network technology, 6TiSCH, provides the ability to detect surrounding information by spraying small dust-sized sensors into physical spaces. There is also a UWB technology that can identify the location of objects indoors through the interactions of surrounding sensors through the wireless sensor network, not through GPS calculation.

### 2.2. RPL

The routing protocol for low-power and lossy networks (IETF) [6] corresponds to a layer 3 protocol in a 6TiSCH network stack. RPL forms a DODAG in a wireless sensor network environment and has a tree-type topology in which a root node acts as a gateway by collecting sensors from each node as a single destination. RPL uses rank to prevent network cycles or loops. Rank represents the relative distance between each node and the root node, and rank calculation depends on the objective function (OF). OF used as a standard in RPL is OF0, and rank is determined based on the expected transmission count (ETX) and received signal strength indication (RSSI). ETX is the estimated number of transmissions and is a measure of the quality of a link between two nodes. The method for calculating this is shown in Equation (1). Slots can be shared or used as dedicated slots for a single node. Shared slots follow the CSMA/CA method. Dedicated slots are formed through a schedule capable of performing one-way communication through negotiation between two nodes. For example, MAC layer broadcast packets are sent using shared slots, and unicast packets are sent through dedicated slots.
Rank(N) = Rank(P) + Rank_increaseRank_increase = (Step) × MinHopRank(1)

The root node that forms a network and relays each node has a rank value of 0. The child nodes of the root node follow Equation (1). Rank(N) is the rank of the current node, and Rank(P) is the rank of the preferred parent node. Rank_increase consists of various parameters.

Step calculates a value from the properties between the parent node, child node, and neighbor. MinHopRank is an index representing the distance between ranks, and the calculation method varies according to the OF.

To form and maintain the topology, the RPL utilizes control messages that are periodically broadcast. The nodes participating in the network to form the topology propagate the DODAG information solicitation (DIS) message to the RPL network. This is a request message requesting a DODAG information object (DIO) message. The node receiving the DIS message broadcasts the DIO. The sensor node receiving the DIO message calculates the OF, selects the node with the lowest rank value as its preferred parent node, and forms a DODAG. The node that calculates the rank sends the information by putting information in the rank item in the DIO message field and propagates its information to neighboring nodes within the communication distance.

The preferred parent node is selected, and the node that forms the DODAG transmits the destination advertisement object (DAO) message from the fixed cell of the TSCH network to the root node, which is the routing table based on the data stored in the DAO message. It stores the address information and forms a routing path. A downlink path can be formed by referring to the DAO message received from the root node and the formed routing table.

Several techniques exist as RPL protocols for mobile nodes. RPL protocols use control messages to form and maintain networks, and there are existing investigations into techniques to improve the trickle timer algorithm, which specifically reduces the broadcast cycle of DIO messages, to avoid conflicts. In [11], the author proposed the trickle-F algorithm. This gives the RPL nodes priority to send DIOs. As a result, the longer the node does not send DIO, the more likely it is to transmit in the next transport frame.

However, mobile nodes may not be able to receive control messages in the process of selecting a parent node in an RPL network by simply moving in any way in the network and improving the trickle timer algorithm, which can increase packet loss and energy consumption. Therefore, in [12], the author proposes an algorithm that dynamically adjusts the transmission cycle of messages based on their own moving speed and direction to select the parent node. This suggests mobility and sends request messages in a timely manner. Because the number of messages used to select the parent node is small, the number of messages required to select the new parent is small. Experiments based on various scenarios revealed that even if the speed and direction of the moving nodes are arbitrary, they perform better than existing algorithms.

A mobility-aware routing protocol based on RPL was presented in EMA-RPL [13]. An EMA-RPL protocol exists as an RPL protocol for moving nodes. The algorithm always defines the mobile node as a leaf in the RPL tree to minimize data loss due to node movement. This also paper defines an associated node (AN) with the mobile node. Data messages generated by the mobile node are sent and received over the AN node, which serves as movement prediction, movement detection, and notification of the mobile node. The mobile node is supported in three ways: mobility detection, response and prediction, and notification. It has the advantage of reducing the energy and resource use of mobile devices through prediction based on the RSSI values of the mobile node and AN node and preventing sending multiple control messages.

A summary of routing protocols for mobile node support in Table 1.

### 2.3. TSCH

IEEE 802.15.4e time slotted channel hopping (TSCH) is one of the media access control (MAC) operating modes of the IEEE 802.15.4e standard protocol for industrial wireless sensor networks such as existing WirelessHART [14] or ISA100.11a [15]. The standard protocol provides high reliability and stability. Through media access control technology that divides time into time slots, and conflicts or contentions that may occur during wireless communication are resolved. In addition, through channel hopping technology, the multipath fading problem can be solved by preventing collisions because of an increase in time slots and a change in frequency. It has high reliability even in a plant environment with many metal structures because of the time-division access characteristic and channel hopping characteristic of hopping multiple channels, and it is possible to guarantee a deterministic delay time by providing communication of devices to the divided time slot according to the schedule.

In the TSCH network, all nodes are synchronized with respect to time. An area composed of a time slot and a channel is called a cell, and each node exchanges packets through the cell. The set of time slots and transmittable channels is referred to as a slot frame, and the slotted frame repeats continuously while the network is being formed. The absolute slot number is the total number of time slots that have elapsed since the network was formed and shared by all nodes. The frequency used by each node in the network was selected based on the equation corresponding to Equation (2). Nc is the total number of channel offsets. The actual frequency f varies for each slot frame because the absolute slot number ASN increases over time.
f = F[(ASN + ChannelOffset) mod Nc](2)

In Figure 2, six nodes(Node A-F) form a TSCH network to exchange data. Packets are sent and received by repeating Tx, Rx, and Sleep in a single cell divided into slot offset and channel offset. In a shared cell, control messages to help form or maintain a network are exchanged. Typical examples are enhanced beacon (EB) [16] for configuring a network or DIO for forming a network topology. Packets contained data can be sent and received in a fixed cell. The scheduling scheme that determines when each node sends and receives packets (in what time slot) and which channel is used is not covered by the IEEE 802.15.4e TSCH standard and varies depending on the specific application. The length of one cell in the TSCH standard is 10 ms, which is the time required to transmit a frame and receive the received ACK. In an unused cell, no packet transmission is performed by any node, and each node maintains a sleep state.

The scheduling technique of TSCH wireless networks depends on the application they choose, and many studies are underway [17]. In addition, there is a centralized technique for the root node, which is the core of the network, to manage all schedules as a central scheduler, or a distributed scheduling scheme that is distributed through the process of negotiating cells between each node. In addition, a link-based scheduling scheme is presented, leveraging the MAC addresses between two different nodes.

A centralized scheduling scheme allows a node to have link information related to the entire network and manages the schedule of the network. The TASA technique proposed in [18] proposes a pattern-and-topology-aware scheduling technique for traffic. However, while these centralized scheduling techniques can theoretically achieve better performance, real-world implementations have the disadvantage of incurring a large number of control messages overhead, resulting in lower scalability of the network.

Distributed scheduling methods provide higher scalability using protocols that negotiate cells between nodes. Therefore, it is an algorithmic method suitable for dynamic or large networks. There is one representative distributed scheduling technique with minimal scheduling function (MSF) presented by six TiSCH WG. This algorithm makes it possible for two neighboring cells to be added/deleted by negotiating 6P to allow two neighboring nodes to add/delete/replace cells in the TSCH schedule. Two adjacent nodes negotiate the location of the cells to be added, deleted, or relocated in the TSCH schedule. This negotiation process is called a 6P transaction, which consists of two or three stages. This scheduling technique does not have an algorithm to support moving nodes and repeats the process of moving nodes to build routing paths.

Ref. [19] proposed BOOST, a protocol that combines TSCH MAC with opportunistic routing and scheduling. Based on layer ID (LID), which is the distance (hop number) from the root node, this protocol determines which channels should be sent or received. The node that wants to send data to the mobile node sends a request message in advance before sending data and sends data packets only to the response node. This achieves 99.99% stability in packet transmission success rate to moving nodes.

The Orchestra scheduling technique [20] is an information-based scheduling method for RPL. Using its own MAC address-based hash function, the slot offset of a dedicated cell is determined without the signal overhead. It features separate slot frames for EB in TSCH, control messages in RPL, and data frames for the independent use of slot frames. It quickly responds to sudden changes in topology without the process of cell negotiation between two nodes, can use RPL to respond to changes in the network, and can take advantage of TSCH’s robustness, reducing end-to-end latency and packet queue length.

Studies exist on scheduling techniques for TSCH standards for moving nodes. In [21], the authors supported node mobility in industrial wireless sensor networks through software-defined network architecture. The architecture of [21] requires a fixed link schedule, and each moving node presents a technique for assigning at least one link to form a fixed link schedule that communicates with each stationary node. However, there is a disadvantage in reducing network scalability by supporting only a limited number of nodes by forming a fixed link schedule for stationary and moving nodes.

The author of [22] proposed MTSCH and introduced the manual enhanced beacon concept to quickly engage mobile nodes in the network. Rather than listening to the EB broadcast in the last time slot of the existing slot offset, it was able to reduce the latency of synchronization messages and save energy compared to the existing TSCH, listening to specific messages only on fixed channels.

In addition to scheduling techniques for the TSCH protocol for moving nodes, there has also been an investigation into the stability of the network with the presence of moving nodes. In [23], the authors analyze the effects of mobility on stability in terms of synchronization, message overhead, latency, and energy. Experimental results show that if the topology configuration of static nodes in the network is not fully covered, the loss during the synchronization process prevents the moving nodes from connecting to the network for a long time, which can cause significant downtime in the network.

In another experiment, we simulate a situation in which the communicable distance of a static node covers the topology of a moving node with all four fixed nodes and one moving node. This generally assumes an industrial environment in which robots move and static gateways ensure stable communication. Experiments show that downtime is extremely low at less than 0.5% regardless of the number of moving nodes, which does not affect TSCH network performance and synchronization if the moving radius of the moving node exists at a distance that can communicate with the static node.

## 3. Efficient Route Management Method for Mobile Nodes in 6TiSCH Network

When a mobile node joins the network to build a routing path, we present a technique for minimizing control message conflicts and the latency from the root node to the root node in response to packets requested by the root node. This technique also minimizes the latency of transmitting responses to packets requested by the moving node from the root node to the moving node through the root node fixed path construction technique. In addition, a technique is presented that allows faster path formation of a moving node by minimizing the conflict of control messages when the moving node disconnects the link to the preferred parent node through the moving node’s fast parent selection technique.

### 3.1. Mobile—Root Node Fixed-Path Construction Techniques

Each node in a 6TiSCH network allocates or removes cells using the 6top protocol after synchronization. If the preferred parent node is changed because of the movement of the moving node, the cell with the previous parent node must be released, and the cell negotiation process with the preferred parent node must go through again. In this process, packet omission occurs, and latency occurs during the process of reallocating cells.

In addition, the moving nodes move while connected to the network and send and receive sensor data with the stationary nodes, which requires the root node to request the data it needs. Therefore, to transfer packets from a moving node to a root node and minimize latency in response to the transmitted packets, we propose a moving-root node down traffic fixed-path construction technique.

The path of mobile root node fixation is formed in three processes. First, when a new node participates in the network and sends a DAO message to the root node, the root node collects the MAC addresses of the nodes participating in the network. We discriminate between stationary and moving nodes by adding a moving flag from the DAO message.

If a DAO message is received from the root node and this DAO message is not a moving node, the slot and channel offsets are stored in automatic cells using the hash value of the MAC address of the node that sent the DAO. This is the basis for assigning a moving node fixed path when forming a moving-root node fixed-path. The root node assigns a move-root node fixed-path initiation cell to a location that minimizes the impact on existing schedules based on the cell information it has.

The root node stores shared and automatic cells, as shown in Figure 3. In Figure 3, a shared cell is a dedicated cell that can send/receive EB, DAO, or DIO control messages, and the alphabet of the automatic cell represents a node participating in the network that is stored in the root node. An automatic cell is a value stored by the root node to allocate the Tx cells of a moving node in a fixed path to the root node, and no communication occurs in that cell.

If we receive a DAO message from the root node and determine from this DAO message that the node currently participating in the network is a moving node, we check the number of cells allocated to each slot offset, regardless of the channel offset. This allows us to assign a dedicated path with the shortest latency from the root node to the moving node. As shown in Figure 4, one cell is allocated for slot offset 1, one cell for slot offset 2, one cell for slot offset 3, one cell for slot offset 4, 0 cells for slot offset 5, one cell for slot offset 6, and 0 cells for slot offset 7. If this is expressed as an array starting with slot offset zero and up to slot offset length, it can be expressed as [1, 1, 1, 1, 1, 0, 1, 0, 0]. The slot offset of 0 was calculated to be 1. This assigns the Tx cell of the root node to the slot offset starting slot with the most consecutive zeros. This requires a move-root node fixed-path to build a packet transmission path that starts with the shortest delay from the root node to the moving node, because each node must be assigned to a continuous slot offset, minimizing the effect of the schedule between the existing stop nodes.

Mobile—allocating Tx cells from the root node in the root node fixed path results in the cell negotiation process of the 6Top protocol from the root node to the moving node. Cells are allocated between root nodes A and B through the cell negotiation process of the 6Top protocol. Node B allows Tx cells on node B to allocate data passed from node E at slot offset 8 where 1 is greater than the Rx cell slot offset of node E, 7, in order to have minimum latency. When the cell negotiation process between nodes E and M is complete, Mobile Node M propagates negotiation completion messages and moving root fixed path routing tables to neighboring nodes in the DIO frame. The DIO is shared with all nodes participating in the network, and the entire node in the network stores the MAC address of the moving-root fixed-path and moving node.

### 3.2. Fast Parent Selection Technique for Mobile Nodes

A technique for generating packets from a moving node to a preferred parent node and forwarding them to the root node has been proposed, but the movement of a moving node often leads to link disconnection from an existing preferred parent node, resulting in packet transmission. If the communication link with the preferred parent node is released, missing packets can have a significant impact on applications that continuously collect data. Therefore, when a moving node recognizes that its link to its preferred parent node is disconnected, it informs the neighboring node that it wants to participate in the topology, receives a DIO message without conflict, and recovers the link quickly.

This technique was performed after the move-root node fixed-path construction technique. While propagating to neighboring nodes in the previously proposed move-to-root node fixed-path, it also delivers the MAC address of the move-to node. At this point, we assign a Tx cell dedicated to unicast control by determining the slot and channel offsets using the agreed hash values of the moving node’s own MAC address and the MAC address of the moving node. The assigned cells remain in a normal sleep state, and when a moving node broadcasts a shared cell with a DIS message stating that it is disconnected from its preferred parent and that it needs to configure a new topology, it sends a unicast DIO message to the moving node. In addition, moving nodes receive DIO messages containing their own MAC addresses propagated by neighboring nodes while maintaining their topology. When a moving node receives a DIO message, it uses the neighboring node and its MAC address to determine the slot and channel offset to assign an Rx cell dedicated to unicast control. Conflicts between neighboring nodes do not occur because they know each other’s MAC addresses and assign cells dedicated to unicast control based on their MAC address hash values.

The cell neighboring node F of the preferred parent node of the current moving node allocates a control-only Tx cell directed at the moving node based on the MAC address with (slot offset, channel offset) = (2, 3). Mobile nodes also allocate control-only Rx cells from neighboring nodes in cells (slot offset, channel offset) = (2, 3). It still communicates with the preferred node; therefore, it assigns a control-only cell, but the radio is off, and hence, packet switching is not performed in that cell.

When a moving node detects a link release from its preferred parent node owing to the movement of a moving node, the moving node broadcasts a DIS message from a shared cell and turns on the control-only cell Rx radio of the moving node. From the following slot frame, a neighboring node can receive unicast control messages from that cell. The neighboring node has an Rx radio turned on in the shared cell and receives DIS messages from the moving node. When a shared cell of a neighboring node detects that it is a DIS message from a moving node, it immediately sends a unicast control message to the moving node from a Tx cell dedicated to unicast control. On a moving node, the radio is turned on in a control-only Rx cell on a neighboring node and unicast control messages are received. The purpose of the technique is to receive DIO messages without conflicts and quickly recover links; therefore, if any of the control-only Rx cells are received, while the radio is off in the control-only Rx cells in the slot offset, they no longer receive unicast control messages from other neighboring nodes.

Figure 5 shows a simplified schedule of mobile nodes in the network shown in Figure 6. The topology shown in Figure 6 is a situation in which the movement of a mobile node releases a link assignment with node E and changes to a preferred parent node with node F. For example, in the first slot frame, in slot frame 7 of mobile node M, there are Rx cells dedicated to control with node D and Tx cells dedicated to data with node E that send and receive data packets.

In the first slot frame, at slot offset 7, the moving node determines that if it fails to transfer data to its current preferred parent node, E, the preferred parent node, the link to the preferred parent node is released and broadcasts a DIS message in a shared cell with slot offset 0 of the next slot frame. To receive unicast control messages from neighboring nodes, turn on and wait for the radio of the unicast control-only Rx cell. A node that receives a DIS message from a shared cell sends a DIO message from a Tx cell dedicated to the unicast control of that slot frame to a moving node and checks the ACK. The moving node sends an ACK when it receives a DIO message from a pre-set unicast control-only Rx cell. The neighboring nodes of the preferred parent node that received the ACK discontinue the task of sending control messages from shared cells. The moving node selects the node with the largest value as the preferred parent node by calculating the incoming signal strength contained in the DIO message received from the unicast control-only Rx sent by each neighbor node. At this point, if the control message is not received from the unicast control-only cell, the back-off technique allows a continuous message transmission to be received. When networking is formed or maintained by the movement of moving nodes, it minimizes DIO message conflicts in shared cells and allows data transfer to be initiated through a fast cell allocation process for moving nodes.

## 4. Performance Analysis

### 4.1. Configuration

To compare the performance of the proposed technique, experiments were conducted using a Python-based 6TiSCH simulator developed by members of the 6TiSCH working group. The 6TiSCH simulator implements the 6TiSCH protocol stack and the trickle timer, as shown in Table 2.

The implementation of the mobility model in the 6TiSCH simulator changes the position of the communication node at the end of each time slot. We assign cells based on the MAC address of each node and use the MAC hash expression used by the minimal scheduling function (MSF) technique to allocate the automatic cells of each node.

### 4.2. Scheduling Performance Analysis

To analyze the performance of the algorithms proposed in this paper, we utilized a tree topology of up to three hop sizes consisting of ten nodes, as shown in Figure 7. In the topology, the distance per node was the same, with a distance of 5 m. A moving node moves at a constant moving speed, such that at least one node in the network is at a communicable distance, generating packets, and sending them back to the root node until they return to the starting point.

The composition of the experimental environment is the first time slot in a slot frame, with six channels in the TSCH network, 10 ms in length in time slots, 21 time slots in slot frames, and one shared cell. One packet was generated per second, and the experiment was conducted at a moving speed that took five s to move from one node to another. Each experiment was repeated 10 times, and the average value of the corresponding results was calculated. The RPL configuration followed the OF0 standard.

#### 4.2.1. Communication Stability Analysis

To compare and analyze the communication stability of the network in which moving nodes exist, the distributed scheduling MSF and Orchestra scheduling techniques proposed by the 6TiSCH working group were compared by implementing the 6TiSCH simulator.

As the first experiment to analyze the communication stability of scheduling, we compare and analyze the latency of each scheduling technique. Figure 8 shows the result of the latency of packets generated by a mobile node to a parent node, and Orchestra, a competitive-based scheduling technique between nodes, has significant latency. Mobile–real-world implementation algorithms with root node fixed paths have the least latency.

#### 4.2.2. Analysis of the Overhead and Resource Usage of Control Packets

Next, an investigation was conducted concerning the overhead and the resource usage of control packets in shared cells. The experiment was analyzed by comparing the number of packets wasted by collision with the control cell to the number of packets wasted by collision through changing the number of nodes at a distance communicable with the moving node in each scheduling technique.

The results of the experiment are shown in Figure 9. Collision in shared cells was the most common in MSF because, after the parent change, it continued to send and receive DIS and DIO control packets and RPL control messages. However, the implemented algorithms minimize the transfer of control messages from the shared cells.

Figure 10 shows a measure of the total number of control packets used to analyze the performance of the scheduling techniques from the resource perspective of control packets. Because the number of parent nodes increased, the number of control packets increased gradually, or approximately 30% more than the MSF algorithm. This is attributed to a greater exchange of control packets per parent node to support the moving nodes.

#### 4.2.3. Stability Analysis of Mobile Nodes

Figure 11 shows the ratio of the total time the moving node is connected to the network and communicable, with Orchestra + EMA-RPL sharing cells for RPL control messages globally; however, cell allocation for data packet communication is automatically determined through MAC hash, allowing participation in the network. However, existing MSF scheduling has a low participation rate because the speed of movement increases, as RPL control messages must compete with existing shutdown nodes through shared cells.

Figure 12 shows the number of packets generated by a moving node that failed to reach the root node as the moving speed increased. Although MSF and Orchestra + EMA-RPL increase rapidly depending on the speed of movement, the implementation algorithm can confirm that the number of packets that failed to transmit is exceedingly small.

We compare Orchestra + EMA-RPL with the distributed scheduling MSF proposed by the existing 6TiSCH working group. Mobile-root node fixed-path construction and fast parent selection of moving nodes are two techniques that show that nodes with a fast moving speed can quickly participate in the network, thus, minimizing missing packets to the root node and delivering less latency.

## 5. Conclusions

In this paper, we proposed a scheduling technique in which a link assignment with a preferred parent node is released owing to the movement of a moving node in a 6TiSCH network. Cell allocation, based on link information from moving and neighboring nodes, quickly resets the path with the preferred parent node and reduces latency.

We compared the scheduling performance of our proposed algorithm with that of Orchestra with EMA-RPL, which supports distributed scheduling and existing moving nodes. Experiments show that when the proposed algorithm is released from link assignment with the preferred parent node because of the movement of the mobile node, it quickly allocates cells through link information from the mobile and neighboring nodes, complementing the path simultaneously, minimizing latency by up to 7× and reducing collision of approximately 2.5× control packets in the shared cell. It was also demonstrated that the proposed algorithm can support mobility better than conventional algorithms by the approximately 33% decrease in network engagement time for mobile nodes.

We focused on algorithms that moved and quickly recovered after the mobile node deviated from the topology. For future research, we plan to extend the study to algorithms with better network quality by registering RSSI values or directional information of mobile nodes as parameters to predict the moving position of mobile nodes in advance.

## Figures and Tables

**Figure 1 sensors-21-03074-f001:**
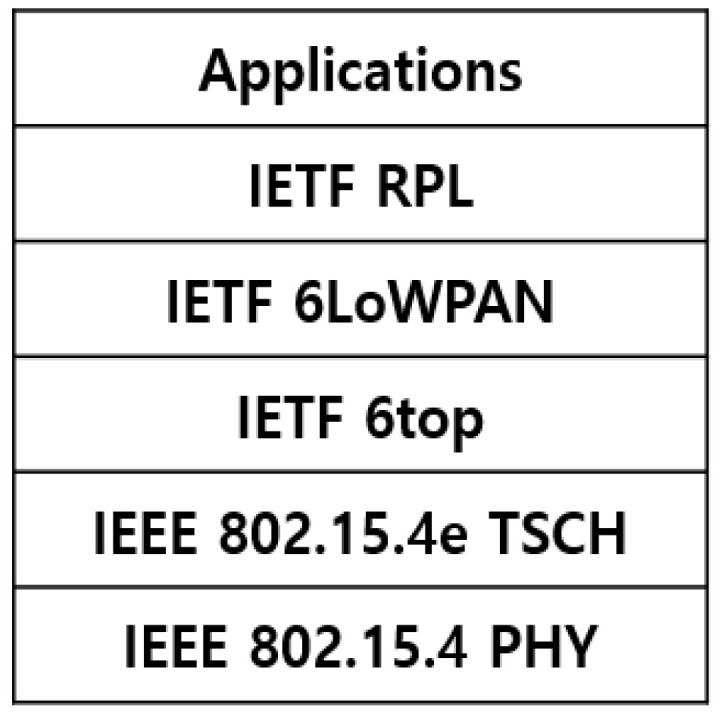
6TiSCH stack structure.

**Figure 2 sensors-21-03074-f002:**
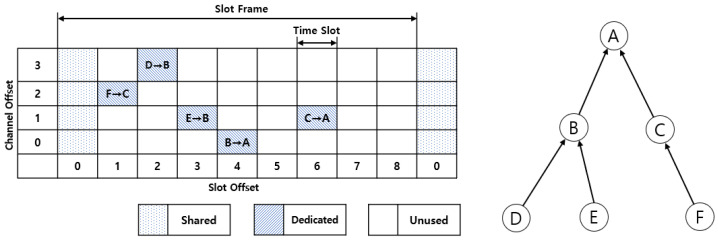
TSCH slot frame and topology.

**Figure 3 sensors-21-03074-f003:**
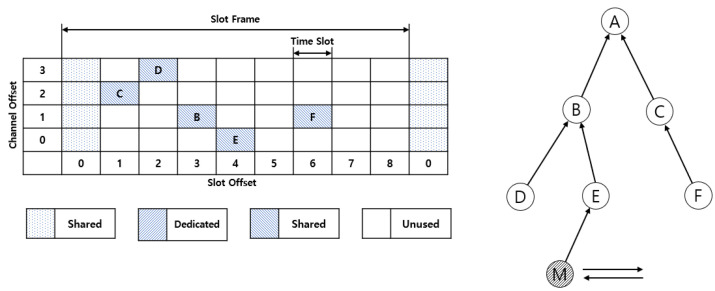
Cell information stored on the root node.

**Figure 4 sensors-21-03074-f004:**
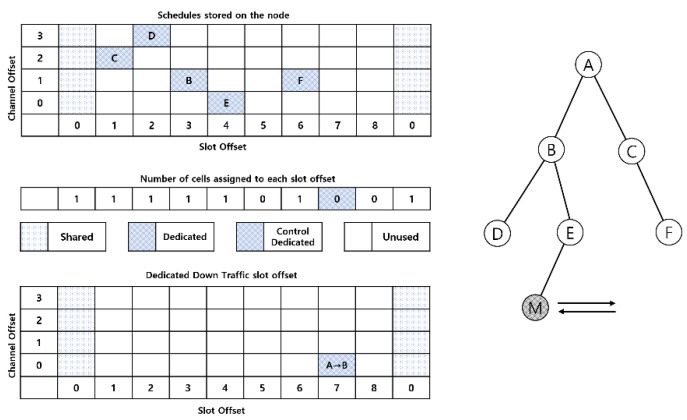
Cell allocation process.

**Figure 5 sensors-21-03074-f005:**
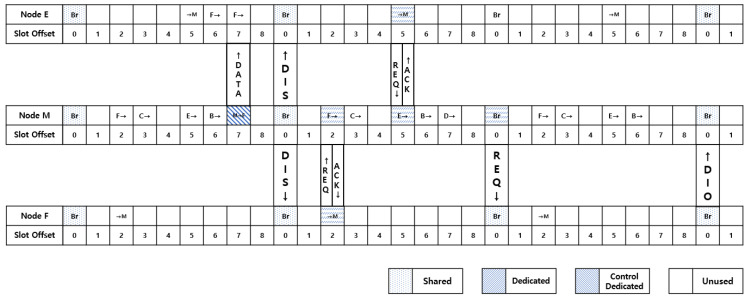
Simplified schedule of nodes that exist at a distance that can be communicated with the mobile node.

**Figure 6 sensors-21-03074-f006:**
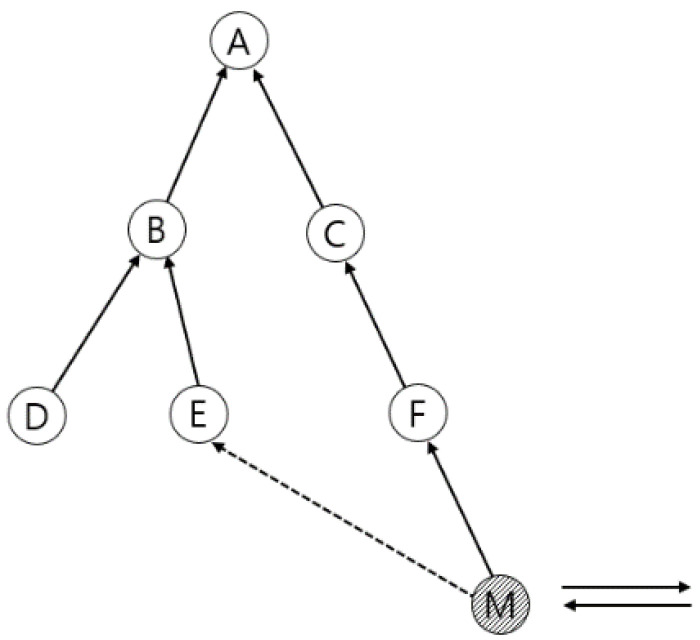
Mobile node changes preferred parent node.

**Figure 7 sensors-21-03074-f007:**
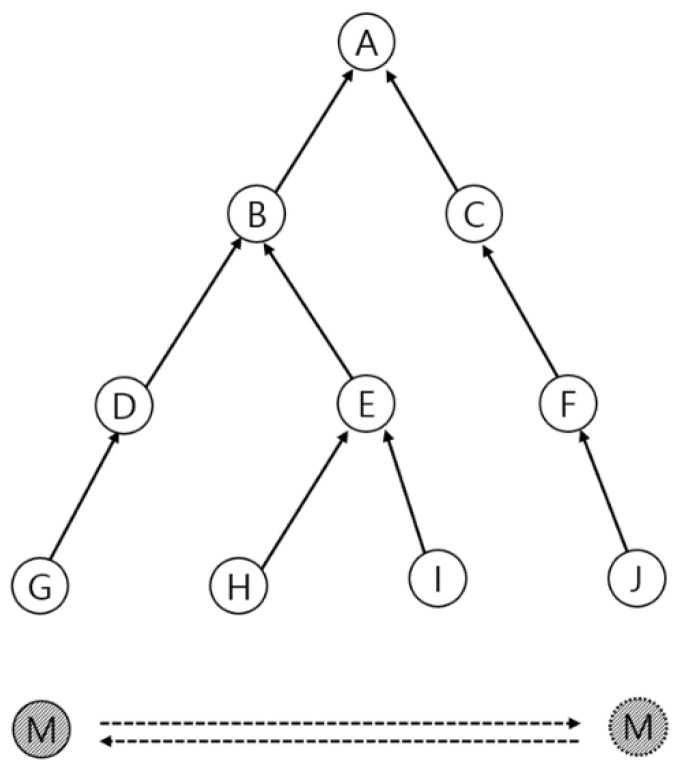
Tree topology with 10 nodes.

**Figure 8 sensors-21-03074-f008:**
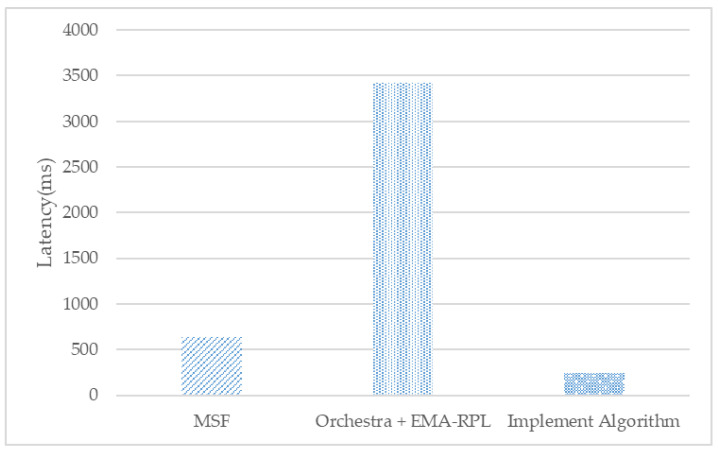
Latency.

**Figure 9 sensors-21-03074-f009:**
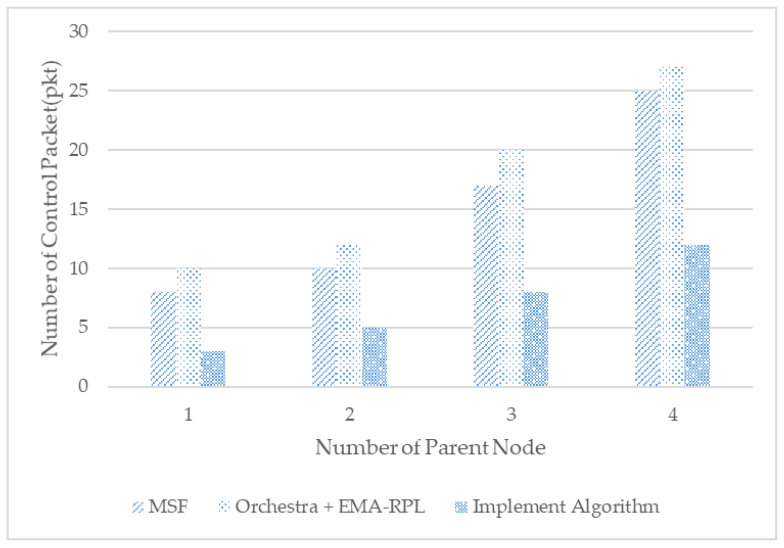
Number of control packets in shared slots.

**Figure 10 sensors-21-03074-f010:**
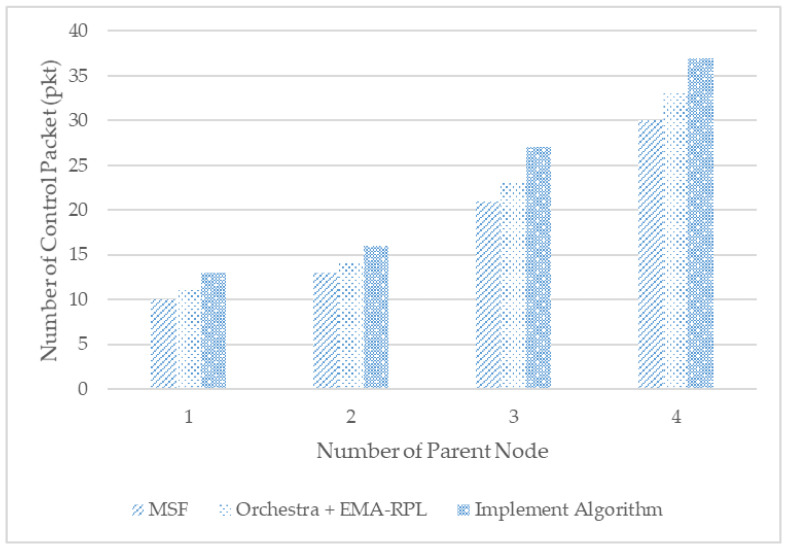
Number of control packets.

**Figure 11 sensors-21-03074-f011:**
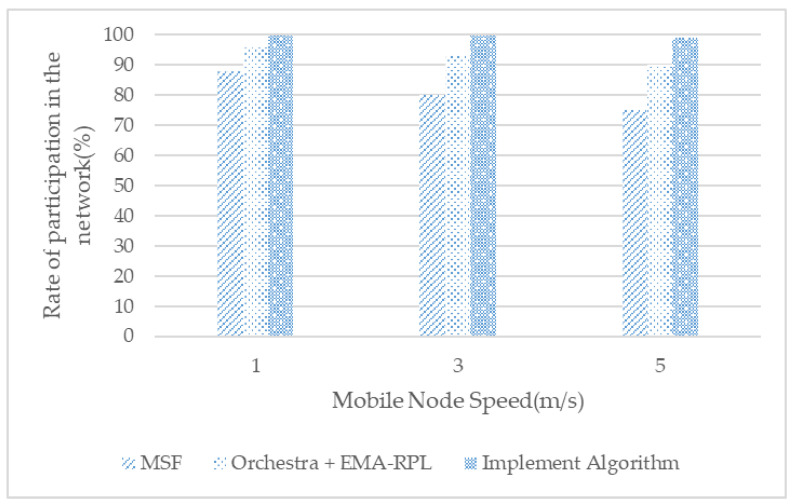
Rate of participating in the network of mobile nodes with node speed.

**Figure 12 sensors-21-03074-f012:**
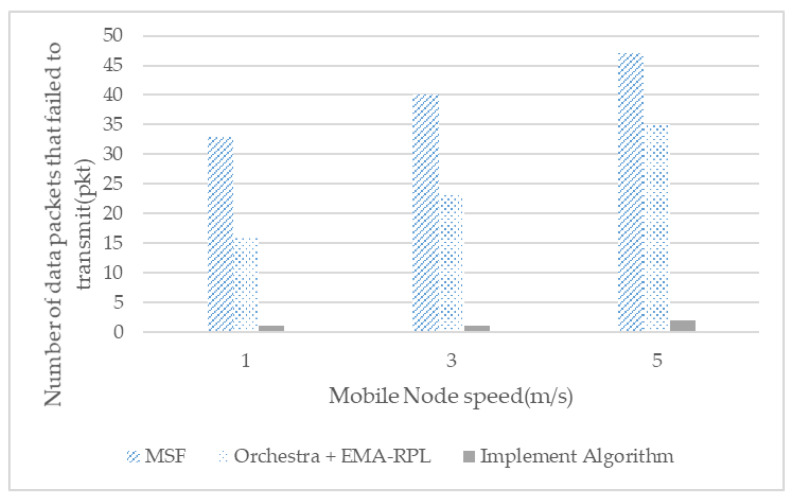
Number of packets that failed to transmit from the mobile node according to the node speed.

**Table 1 sensors-21-03074-t001:** A summary of routing protocols for mobile node support.

Proposal	Description
Trickle-F [11]	The RPL nodes are prioritized to transmit DIOs to adjust the transmission cycle.
Algorithm for Energy-Efficient Node Mobility [12]	Dynamically adjust the transmission interval of the message according to the speed and direction of travel. As mobile nodes move, fewer messages are required per unit hour to select a new parent node.
EMA-RPL [13]	Neighboring nodes of the moving node predict the movement of the moving node through the RSSI value and select the parent node.

**Table 2 sensors-21-03074-t002:** 6TiSCH simulator protocol stack

RFC6550, RFC6552	RPL, non-storing mode, OF0
RFC6206	Trickle Algorithm
draft-ietf-6lo-minimal-fragment-07	6LoWPAN Fragment Forwarding
RFC6282, RFC4944	6LoWPAN Fragmentation
draft-ietf-6tisch-msf-10	6TiSCH Minimal Scheduling Function (MSF)
draft-ietf-6tisch-minimal-security-15	Constrained Join Protocol (CoJP) for 6TiSCH
RFC8480	6TiSCH 6top Protocol (6P)
RFC8180	Minimal 6TiSCH Configuration
IEEE802.15.4-2015	IEEE802.15.4 TSCH

## Data Availability

All the data generated or analyzed during the current study are included in the published article.

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
