# Peer review of "Efficient Route Management Method for Mobile Nodes in 6TiSCH Network"

_sensors, 2021, doi:10.3390/s21093074_

Round 1

Reviewer 1 Report

The article "Efficient Route Management Method for Mobile Nodes in
6TiSCH Network "is interesting for the magazine and if improved it can be published and have contributions to readers. However, some points need to be improved:

Introduction
It is necessary to review some points, mainly related to the first citation of a reference with more than 10 years related to IoT. There are recent works with innovative and current solutions, and also this one that presents innovative solutions in the context of adaptation in IoT environments: https: //doi.org/10.3390/computers9040077

Section 2
It needs to be improved, it is necessary to compare it with recent works and insert a comparative table between the work developed and the others compared.
New technologies that could be used even as an application scenario could be included for greater understanding of the reader, e.g. deals with communication protocols used; work that involves solutions with Blockchain.

Figures
It is imperative that the quality of the images, figures is improved. Reconsider the number of figures 20, reevaluate keeping a maximum of 10 and the rest as appendix.
Even the figures that remain should be better detailed.

Conclusions
Dear authors, the conclusions of the work are summarized in 13 lines. It is extremely relevant that you value your work developed in this section. They insert data and contributions obtained, results, discussions that motivate the reader about the development of the work done. I suggest that they briefly describe future work, which can still be done to improve the work developed.

I suggest including a list of Abbreviations, there are several abbreviations used in the text.

Author Response

Response to Reviewer 1 Comments

Thank you very much for your interest and time in our paper. I revised the part that the reviewer said for a better paper. If there is a lack of modification, I will try to fix it additionally.

The following is a sequence of what the reviewer suggested and the responses to that part.

The article "Efficient Route Management Method for Mobile Nodes in

6TiSCH Network "is interesting for the magazine and if improved it can be published and have contributions to readers. However, some points need to be improved:

Introduction

It is necessary to review some points, mainly related to the first citation of a reference with more than 10 years related to IoT. There are recent works with innovative and current solutions, and also this one that presents innovative solutions in the context of adaptation in IoT environments: https: //doi.org/10.3390/computers9040077

Response Introduction: I agree with the reviewer's opinion that references to older IoT over 10 years appear. We added a paper on IoT that is related to the current solution.

Section 2

It needs to be improved, it is necessary to compare it with recent works and insert a comparative table between the work developed and the others compared.

New technologies that could be used even as an application scenario could be included for greater understanding of the reader, e.g. deals with communication protocols used; work that involves solutions with Blockchain.

Response Section 2: We added a survey of related studies and A summary of routing Protocol for mobile nodes' support table.

Figures

It is imperative that the quality of the images, figures is improved. Reconsider the number of figures 20, reevaluate keeping a maximum of 10 and the rest as appendix.

Even the figures that remain should be better detailed.

Response Figure: We agree with Figure's revision of the number of figures, but We think it is necessary to understand my paper. The data after Figure 14 is about the results of the experiment.

Conclusions

Dear authors, the conclusions of the work are summarized in 13 lines. It is extremely relevant that you value your work developed in this section. They insert data and contributions obtained, results, discussions that motivate the reader about the development of the work done. I suggest that they briefly describe future work, which can still be done to improve the work developed.

Response Conclusions: Thank you for your great suggestion. I added a part about the future work to the conclusion section and added a part about my experiment.

I suggest including a list of Abbreviations, there are several abbreviations used in the text.

Response list of Abbreviations: After the Conclusion section, you added the list of Abortion section.

Reviewer 2 Report

As the title indicates, the authors proposed route management method for mobile nodes in 6TiSCH network. The topic of this paper seems relevant to the scope of journal. But, the paper cannot be considered for the acceptance.

1) The English should be improved significanlty. It is recommended to use English editing service before re-submission.

2) When it comes to the references, current version is under standdard. Basically, it is required to use the same format for journals and conferences. Also, references should be updated for state-of-the-arts in this research area.

3) The presentation of figure and table is not acceptable. 
4)Typos and errors such as "TSCH's time slot" -> time slot in TSCH, "Owing to the characteristics of TSCH and RPL designed based on a static node, there is a problem that the network cannot respond quickly to recover by exchanging contention-based control packets in a shared cell when the mobile node reconnects to the network because of the node’s movement." -> Too long sentences, 

"in a shared cell when the mobile node reconnects to the network" -> duplicated sentences in abstract, 

"In this paper, we propose a scheduling" -> past sentence, we proposed

"We compare the scheduling performance" -> past sentence, we compared

5) Analysis for simulation results is strongly improved to give a confidence for the proposed scheme.
6) Another imporant thing for the reviewer 3 comment, that is, Accepted in Current form

"Dear Authors,

The submitted paper “Replication Study Evaluating SNS Continuance in the South Korean Context" is addressing an important and interesting topic, therefore thank you very much for your work and the contribution".

I think the reviewer 3 point out the other paper comment now. It should be corrected.

7)  In the conclusion, it is usual to use "Past tense". In this paper, we propose a scheduling -> we proposed

Author Response

Response to Reviewer 2 Comments

Thank you very much for your interest and time in our paper. I revised the part that the reviewer said for a better paper. If there is a lack of modification, I will try to fix it additionally.

The following is a sequence of what the reviewer suggested and the responses to that part.

As the title indicates, the authors proposed route management method for mobile nodes in 6TiSCH network. The topic of this paper seems relevant to the scope of journal. But, the paper cannot be considered for the acceptance.

Point 1: The English should be improved significanlty. It is recommended to use English editing service before re-submission.

Response 1: We are submitting after modifying grammatical parts and awkward phrases. Thank you for pointing it out.

Point 2: When it comes to the references, current version is under standdard. Basically, it is required to use the same format for journals and conferences. Also, references should be updated for state-of-the-arts in this research area.

Response 2: The section on Standard has been modified to refer to journals or conference materials. Thank you very much for telling me what you didn't know.

Point 3: Typos and errors such as "TSCH's time slot" -> time slot in TSCH, "Owing to the characteristics of TSCH and RPL designed based on a static node, there is a problem that the network cannot respond quickly to recover by exchanging contention-based control packets in a shared cell when the mobile node reconnects to the network because of the node’s movement." -> Too long sentences,

"in a shared cell when the mobile node reconnects to the network" -> duplicated sentences in abstract,

"In this paper, we propose a scheduling" -> past sentence, we proposed

"We compare the scheduling performance" -> past sentence, we compared

Response 3: Corrected grammar mistakes for Abstract. I wrote them all using the past tense. Thank you for pointing it out.

Point 4: In the conclusion, it is usual to use "Past tense". In this paper, we propose a scheduling -> we proposed

Response 4: We've filled it out using the past tense for conclusion.

Reviewer 3 Report

Dear Authors,

The submitted paper is addressing an important and interesting topic, therefore thank you very much for your work and the contribution.

The paper focuses on the combination of TSCH's time slot access and multi-channel and channel hopping  technology in industrial IoT sensor networks. It is interesting research area as it enables high reliability and low power operation to meet the stability and real-time requirements of industrial applications. The scheme proposed by authors minimizes the collision of control messages by allocating a control-only cell to form a preferred parent node, Additionally the authors introduced a method for establishing a fixed route from the root node to the mobile node to minimize the delay time. Superiority of authors’ approaches was confirmed in simulation environment.

From a methodical and practical points of view the paper scores high. Results are very interesting and give strong contribution in the subject under investigation. Paper is well written. Literature is relevant. Generally, paper is well structured, important theoretical and  practical aspects of the examined problem are studied and presented in a clear and consistent manner.

To sum up - paper interesting and technically sound. Thus I suggest to accept the paper.

As a suggestion - please extend the relevant and up to date literature 

Author Response

Response to Reviewer 3 Comments

Thank you very much for your interest and time in our paper. I revised the part that the reviewer said for a better paper. If there is a lack of modification, I will try to fix it additionally.

We have updated on IoT-related references and added Table to Section2 to make them easier to understand.

Round 2

Reviewer 1 Report

Dear authors, several requests were not made, suggestions pointed out in the previous review. My role as a reviewer is mainly to try to help improve the quality of the article produced. Therefore, it is assumed that in addition to what was suggested, the review also has other improvements, however not even what was initially requested was accomplished or justified.

Author Response

Thank you very much for your interest and time in our paper.
Here are the corrections you pointed out.

- Edit in the abstract, integration, and conclusion section.
- Added the latest solutions to IoT and industrial wireless network protocols.
- Revised Figures.
- Added results and future research to the Conclusion Section.

Reviewer 2 Report

Even though the authors have addressed the comments, English should be improved for camera-ready paper. Also, final careful editing is stongly required.

Author Response

Thank you very much for your interest and time in our paper.
Here are the corrections you pointed out.

- Edit in the abstract, integration, and conclusion section.
- Added the latest solutions to IoT and industrial wireless network protocols.
